# Systematic Review of Calcium Channels and Intracellular Calcium Signaling: Relevance to Pesticide Neurotoxicity

**DOI:** 10.3390/ijms222413376

**Published:** 2021-12-13

**Authors:** Carmen Costas-Ferreira, Lilian R. F. Faro

**Affiliations:** Departamento de Biología Funcional y Ciencias de la Salud, Facultad de Biología, Universidade de Vigo, Campus Universitario As Lagoas Marcosende, 36310 Vigo, Spain; maica.cf@hotmail.com

**Keywords:** Ca^2+^ channels, Ca^2+^ homeostasis, Ca^2+^ binding proteins, pesticides, nervous system

## Abstract

Pesticides of different chemical classes exert their toxic effects on the nervous system by acting on the different regulatory mechanisms of calcium (Ca^2+^) homeostasis. Pesticides have been shown to alter Ca^2+^ homeostasis, mainly by increasing its intracellular concentration above physiological levels. The pesticide-induced Ca^2+^ overload occurs through two main mechanisms: the entry of Ca^2+^ from the extracellular medium through the different types of Ca^2+^ channels present in the plasma membrane or its release into the cytoplasm from intracellular stocks, mainly from the endoplasmic reticulum. It has also been observed that intracellular increases in the Ca^2+^ concentrations are maintained over time, because pesticides inhibit the enzymes involved in reducing its levels. Thus, the alteration of Ca^2+^ levels can lead to the activation of various signaling pathways that generate oxidative stress, neuroinflammation and, finally, neuronal death. In this review, we also discuss some proposed strategies to counteract the detrimental effects of pesticides on Ca^2+^ homeostasis.

## 1. Introduction

Calcium (Ca^2+^) is a critical second messenger in almost all cell types. This ion plays a central role in the nervous system, where it is closely linked to the regulation of numerous neuronal functions such as neurotransmission, neuronal excitability, or gene expression [1,2]. Likewise, Ca^2+^ also participates in long-term processes such as learning and memory [3].

Under physiological conditions, the intracellular concentration of Ca^2+^ ([Ca^2+^]i) in unstimulated cells is kept between 50 and 100 nM, which implies the need to ensure a marked balance between the influx and efflux of this ion, achieved by the combined action of transporters, channels, pumps, exchangers, and buffers (Figure 1). Consequently, increasing its influx from the extracellular medium or its release from intracellular organelles can raise its concentration in the intracellular environment. A small proportion of this Ca^2+^ binds to the different effectors that are responsible for the activation of the many Ca^2+^-dependent pathways within the cell (Figure 1). The stimulation of these pathways can alter numerous intracellular processes, such as neurotransmitter release, gene expression, or cytoskeletal histoarchitecture, essential for cell function [4,5,6]. On the other hand, an excessive increase in Ca^2+^ levels can also activate a series of harmful mechanisms for the cell, such as the alteration of mitochondrial functioning and the generation of free radicals, which can finally cause cell death by apoptosis [7,8].

The increase in the intracellular Ca^2+^ levels can take place through two main mechanisms: the entry of Ca^2+^ from the extracellular space and/or its release from internal stores [9,10]. Regarding Ca^2+^ entry, this can occur through various types of channels and receptors present in the plasma membrane, including voltage-gated calcium channels (VGCC), receptor-operated channels (ROC), or storage-operated calcium channels (SOC) [11]. Regarding the release of Ca^2+^ from internal compartments, this ion can come mainly from the endoplasmic reticulum (ER) or the mitochondria [12,13].

Abnormal Ca^2+^ homeostasis has been consistently associated with the etiopathogenesis of various diseases of the central nervous system (CNS) and neuropathy caused by exposure to various types of toxic substances [14,15,16]. Likewise, recent findings indicate that the development of some neurodegenerative diseases could be closely related to the selective vulnerability of some populations of neurons to Ca^2+^ dyshomeostasis [17,18,19]. These data indicate that the CNS appears to be especially vulnerable to damage caused by disturbances in intracellular Ca^2+^ concentrations.

The term pesticide refers to any substance or mixture of substances of an organic or inorganic nature intended to combat unwanted organisms such as insects, rodents, fungi and even plants, which can be pathogens and/or interfere with production, processing, storage, transportation, or the marketing of food or other articles. In recent decades, numerous studies have focused on the potential of many pesticides to cause adverse health and environmental effects. Thus, exposure to these compounds is associated with deleterious effects on reproduction (such as infertility or congenital malformations), induction of cancer (mainly leukemias and lymphomas), possible alterations in the immune and endocrine systems, and neurological implications, with alterations in the behavior, the appearance of polyneuropathies and the development of neurodegenerative diseases such as Parkinson’s and Alzheimer’s diseases and Huntington’s chorea [20,21,22].

Pesticides can be broadly classified in different ways, being divided into groups or classes according to certain similarity criteria that they share. These different classifications make it easier to identify the potential risks they present. Thus, the classification recommended by the World Health Organization [23] is the one that classifies pesticides according to their acute toxicity on rats as extremely hazardous (Ia), highly hazardous (Ib), moderately hazardous (II), slightly hazardous (III), and unlikely to present acute hazard (U). Other classifications consider the target pest species (insecticides, fungicides, rodenticides, etc.), their origin (organic, inorganic, biological), or their function (repellents, fumigants, feeding deterrents, etc.). However, the most used classification is the one that considers the chemical group (Figure 2).

The extensive use of pesticides constitutes a serious public health problem worldwide, causing approximately 300,000 deaths each year, with developing countries being the most affected [24]. The industrialization of the agricultural sector has led to a very significant increase in the use of pesticides aimed at improving crop yields. In addition, it is estimated that in the coming years the amount of pesticides discharged into the environment will be even greater, due to the need to provide food to a growing world population, as well as to deal with various infectious diseases [25]. Therefore, all human populations are inevitably exposed to this class of chemicals, either directly or indirectly.

Direct exposure to pesticides occurs during manufacturing, transportation, sale and application processes in crops, and also occupationally [25]. Data available in the literature show that this exposure to high concentrations of pesticides in agricultural environments has a direct negative impact on human health [26,27]. Furthermore, children who live in agricultural settings and/or whose parents are occupationally exposed to these compounds are especially vulnerable to their toxic effects [25,28].

On the other hand, indirect or non-occupational exposure includes the ingestion of pesticide residues in contaminated food and water, as well as the inhalation of pesticide droplets present in the air [29]. Although in most cases the concentrations present in food do not exceed the safety limits determined by legislation, in several regions around the world the pesticide residues detected in food or in water clearly exceed the permitted limits [30,31,32]. Furthermore, it should also be considered that the safety limits established for individual pesticides may underestimate the true risk to health, since in real life people are chronically exposed to pesticide mixtures, the toxic effects of which may be independent, additive, or synergistic [33]. Other important risk factors are the persistence in the environment and the ability to bioaccumulate and biomagnify within the food chain, further increasing its potential toxic risk [25]. These data reflect the complex nature of the problem, as chronic exposure to different combinations of pesticides could cause adverse health effects not yet known.

In the CNS, exposure to different classes of pesticides has been related to a series of neurotoxic effects produced through different mechanisms of action, such as alterations in neurotransmission, oxidative stress, or neuroinflammation, which can result in neuronal death. Intracellular Ca^2+^ homeostasis may also be an important target for the manifestation of morphological changes and cellular dysfunction that occur because of pesticide exposure [34,35,36]. These alterations in Ca^2+^ homeostasis can occur through various mechanisms, such as the action of pesticides on the different types of Ca^2+^ channels of both vertebrates and insects [37], in addition to alterations in the functioning of calcium-binding proteins or other intracellular signaling pathways.

In recent years, there has been a significant increase in the number of scientific publications attempting to identify the specific mechanisms by which pesticides disrupt the mechanisms that regulate intracellular Ca^2+^ levels, Ca^2+^ channels, and intracellular Ca^2+^-dependent signaling pathways. Nevertheless, a full review of the research articles on the effects of pesticides on the Ca^2+^ homeostasis in the nervous system is still needed. Therefore, the objective of the present systematic review is to identify the mechanisms of action and the effects that different classes of pesticides exert on the Ca^2+^ intracellular homeostasis in the CNS of the vertebrates and insects. Given the close relationship between the alteration of intracellular Ca^2+^ levels and neurodegeneration, the final purpose of this work is to know the risks derived from increasing exposure to pesticides present in the environment and in food.

## 2. Methodology

The present review was carried out with the aim of unifying the main results of studies on the effects of different classes of pesticides on calcium homeostasis in the nervous system. For this purpose, a systemic review was performed following the guidelines established by Preferred Reporting Items for Systematic Reviews and Meta-Analyses (PRISMA) [38]. Searches were carried out in the specialized databases PubMed, Scopus, and Web of Science in September 2021, with a time restriction limited to studies published in the last twenty years.

To identify the scientific articles to be included in this manuscript, several terms were included in the databases according to the following search strategy: “((calcium homeostasis) AND (nervous system) AND (pesticide))”.

### Inclusion and Exclusion Criteria

The articles included in this review met the following inclusion criteria: (1) original studies in article format; (2) in English or Spanish; (3) studying the effects of pesticides on calcium homeostasis; (4) studying the effect on the nervous system. Studies were excluded based on the following exclusion criteria: (1) theoretical articles or reviews; and (2) studies evaluating the effects of substances other than pesticides.

As a result of the searches carried out in the three databases, a total of 251 articles published in the last 20 years were identified. The articles were then exported to RefWorks and those that were duplicated were removed. According to the inclusion and exclusion criteria, we screened the 239 titles and abstracts to verify compliance with the inclusion and exclusion criteria. After this procedure, 195 articles were excluded for the reasons described in Figure 3. Finally, 28 studies were selected to form part of the present review.

## 3. Results

The studies included in the present review show that exposure to various classes of pesticides produce significant alterations in the intracellular Ca^2+^ homeostasis, both in neurons and in glial cells. Most of the data analyzed are from in vitro studies carried out with cell cultures from both vertebrate and invertebrate animals.

### 3.1. General Effects of Pesticides on Ca^2+^ Homeostasis

As previously discussed, Ca^2+^ is a second messenger that plays a central role in the regulation of numerous biological processes. Maintaining Ca^2+^ levels within physiological limits is a necessity since the increased influx of this ion constitutes an important mechanism of toxic cell death. In line with this, it has been widely documented that various organophosphate, pyrethroid, and carbamate pesticides, as well as rotenone, were able to significantly increase basal intracellular Ca^2+^ levels [37,39,40,41,42,43,44,45,46,47,48,49,50]. The main results of these studies are shown in Table 1. The increase in intracellular Ca^2+^ concentration induced by pesticides had already been documented in previous studies with various cell lines [14,51,52,53].

There are numerous potential toxic effects that these pesticide-induced increases in [Ca^2+^]i can cause to the nervous system, which put the survival of non-target organisms at risk. Such is the case of the insecticide flubendiamide, whose application induced strong Ca^2+^ transients in the antennal neurons of honeybees [56]. Because of these alterations, it is possible that antennal neurons incorrectly detect and encode chemical signals, which can cause a disruption in the social dynamics necessary for the survival of this species. Similarly, the team of Martin et al. [57] observed in *Drosophila melanogaster* that exposure to ziram caused spontaneous and synchronized bursts of Ca^2+^ entry in Type II aminergic nerve terminals, but not in Type Ib glutamatergic nerve terminals. These results suggest the specific sensitivity of aminergic neurons to the pesticide and the implications this could have for the development of pathologies in which the aminergic systems are severely compromised, such as Parkinson’s disease.

However, the changes in the [Ca^2+^]i induced by some organophosphates were substantially different. On the one hand, the team of Qian et al. [55] demonstrated that exposure to different concentrations of paraoxon (1–30 μM) caused a significant decrease in [Ca^2+^]i in CCF human astrocytoma cells, an effect that was not observed for the DFP insecticide. In line with these results, in the research by Vatanparast et al. [59] it was found that paraoxon (0.3 μM) caused a decrease in the amplitude and duration of the Ca^2+^ action potentials, and downregulated the immediate post-hyperpolarization, thus increasing the cellular activity. Nevertheless, in a later study, Vatanparast et al. [58] found that a higher dose of paraoxon (0.6 μM) also reduced the duration of action potentials, but increased the duration of post-hyperpolarization, which caused a reduction in neuronal excitability. Similarly, Hong et al. [46] observed that the addition of a high concentration of paraoxon (2 mM) to SH-SY5Y cells caused a transient increase in the [Ca^2+^]i, while repeated treatment with a reduced dose of pesticide (0.05 mM) decreased the levels of the ion within the differentiated cells. These data suggest that paraoxon could exert a different effect on the [Ca^2+^]i in a dose-dependent manner.

Meijer et al. [54] also found that in vitro exposure to micromolar concentrations of organophosphate pesticides (0.1–10 μM) inhibited, in a concentration-dependent manner, the increase in the [Ca^2+^]i caused by depolarization. This result suggests that the inhibition of Ca^2+^ influx evoked by depolarization could be specific to organophosphate since this effect was not observed after exposure to carbaryl, a carbamate insecticide [54].

### 3.2. Effects on Ca^2+^ Channels and Plasma Membrane Ca^2+^ Pumps

The effects observed on the functioning of membrane channels and pumps are shown in Table 2. As commented above, one of the main mechanisms leading to an increase in [Ca^2+^]i is by its influx through channels present in the plasma membrane. There is a wide variety of channels that allow Ca^2+^ influx, including VGCCs, SOCs and ROCs, with their open or closed state markedly influencing the [Ca^2+^]i in neurons.

VGCCs are present in many cell types, where they are activated during membrane depolarization and mediate the Ca^2+^ influx necessary to initiate synaptic transmission at conventional synapses [66]. There are several types of VGCCs with distinct electrophysiological and pharmacological properties. Based on the conductance of these channels, several currents were identified and named: L-type, T-type, N-type, P/Q-type, and R-type [67].

It has been shown that exposure to different classes of pesticides can affect the proper functioning of VGCCs. Several in vivo and in vitro studies have found that carbofuran, dichlorvos, deltamethrin, methamidophos, and rotenone exert their action on VGCCs, increasing Ca^2+^ influx through these channels [37,39,42,47,50,60,65]. Specifically, it appears that the increases in [Ca^2+^]i induced by rotenone, dichlorvos and methamidophos depend on their action on L-type VGCCs [39,42,65], whereas the effect of deltamethrin on Ca^2+^ homeostasis could be mediated by L- and T-type VGCCs [37,60]. Likewise, other in vivo studies have also suggested the mediating effect of L-type VGCCs on the enhanced Ca^2+^ influx induced by organophosphates [14,68]. 

In contrast, the in vitro study by Meijer et al. [62] observed that exposure to low concentrations (0.1–100 μM) of the organochlorine endosulfan, the organophosphate chlorpyrifos, or the pyrethroid cypermethrin inhibited depolarization-induced Ca^2+^ influx under subchronic conditions. This effect was attributed to slow or non-reversible inhibition of VGCCs. In this study, Meijer et al. [62] also found that repeated exposure to chlorpyrifos increased the inhibition of VGCCs compared to single exposure. However, this effect was not observed after repeated exposure to chlorpyrifos-oxon, cypermethrin, or endosulfan. In line with this, it has been shown that the parent compounds of organophosphates are more potent than their -oxon metabolites in blocking VGCCs and thereby inhibiting the depolarization-induced increase in [Ca^2+^]i [54]. Therefore, according to the data of Meijer et al. [62], the metabolization of organophosphates would imply a partial bioinactivation/detoxification of these compounds since their metabolites have a lesser effect on the activity of VGCCs. This finding is especially relevant for risk assessment of organophosphate exposure in humans, since young animals and humans exhibit a lower rate of pesticide metabolism [69]. Thus, the parent compounds could remain longer in the body and exert a significant neurotoxic effect during development.

Another important finding from the research of Meijer et al. [54] was that exposure to binary mixtures of chlorpyrifos with its analog -oxon or with parathion-ethyl did not increase the degree of inhibition on the [Ca^2+^]i increase elicited by depolarization. These non-additive effects contrast with previous observations of other classes of pesticides, in which mixtures of organochlorine insecticides or azole fungicides exerted additive inhibition of VGCCs [70,71]. The non-existence of an additive interaction between organophosphates could be explained by their possible competition for different VGCCs or by the potential interaction between this class of compounds, which would lead to a decrease in the levels of the active substance interacting with VGCCs. However, there is still not enough information available to be able to draw rigorous conclusions and further studies are needed to clarify the absence of additive effects for organophosphates.

On the other hand, the increase in cytosolic Ca^2+^ mediated by VGCCs during the action potential can induce a number of physiological processes, such as the activation of [Ca^2+^]i-dependent K^+^ currents, which contribute to repolarization and after-potential hyperpolarization [72]. The hyperpolarization following action potentials constitutes an important intrinsic negative feedback mechanism that regulates neuronal excitability. Thus, a decrease in the duration of hyperpolarization will increase cell activation.

In line with the above, findings from the studies of Vatanparast et al. [58,59] suggest that paraoxon could suppress Ca^2+^ entry during the action potential by blocking VGCCs. The results suggest that exposure to this pesticide could trigger some adaptive mechanisms to regulate the functionality and availability of VGCCs [73]. Blockade of VGCCs would negatively regulate small conductance of Ca^2+^-dependent K^+^ channels, causing a reduction in downstream hyperpolarization and a consequent increase in neuronal activation. Similar findings have been reported in previous studies, in which exposition to soman and VX reduced the duration of hyperpolarization by decreasing Ca^2+^ entry during the action potential [74,75]. In general, increased neuronal excitability has been found to be a common effect of acute organophosphate poisoning [59].

Activation of SOCs constitutes another important pathway for the increase in [Ca^2+^]i in virtually all cells. SOCs are not voltage-dependent but are activated by depletion of ER Ca^2+^ stores with the goal of allowing faster return of the ion to the intracellular stores [76]. Physiologically, SOCs are activated by stimuli that release Ca^2+^ from the ER, which generally involves activation of inositol 1,4,5-trisphosphate (IP_3_) or ryanodine channels/receptors [77]. These Ca^2+^ channels have been found to be involved in increases in [Ca^2+^]i induced by both malathion and the pyrethroid lambda-cyhalothrin [44,45].

Another type of channel present in the neuronal membrane is the transient receptor potential channel (TRP channel), a type of non-selective cation channel that allows Ca^2+^ entry and plays a fundamental role in sensory signaling [78]. Within this family is the transient receptor potential cation channel subfamily A member 1 (TRPA1), expressed in nociceptive neurons of the dorsal root ganglion and trigeminal ganglion. This channel is activated by a variety of stimuli, including spicy compounds and environmental irritants [79]. The findings of the study by Ding et al. [40] suggest that malathion could favor the Ca^2+^ entry through TRPA1, thereby increasing its levels inside neurons. Another member of this family is the transient receptor potential cation channel subfamily M member 2 (TRPM2), which is widely expressed in various regions of the CNS [80]. This channel allows Ca^2+^ entry into the cell and is activated mainly by intracellular ADP-ribose and Ca^2+^, as well as by reactive oxygen species (ROS) and reactive nitrogen species [81]. In line with this, the team of Freestone et al. [43] found that the rotenone-induced increase in [Ca^2+^]i in substantia nigra *pars compacta* neurons was dependent on TRPM2 activation.

Another mechanism involved in the pesticide-induced dysregulation of Ca^2+^ homeostasis could be the activation of some ROCs present in the cell membrane, both ionotropic and metabotropic. Thus, it has been shown that muscarinic (mAChRs) and nicotinic (nAChRs) acetylcholine receptors could be one of the molecular targets of organophosphates pesticides. The neurotoxicity exerted by organophosphates is mainly due to the irreversible inhibition of acetylcholinesterase activity, resulting in the accumulation of acetylcholine in the synaptic cleft and the subsequent activation of their respective receptors [82]. However, it has also been shown that organophosphates can bind to mAChRs and nAChRs, acting either as agonists or as inhibitors, depending on the organophosphate, receptor subtype, and cell type [46,83,84,85,86,87,88,89]. This modulation of cholinergic channels/receptors affects the [Ca^2+^]i, since the activation of nAChRs induces the direct entry of Ca^2+^, while the stimulation of mAChRs can trigger signal transduction cascades leading to phosphorylation/dephosphorylation of Ca^2+^ channels and/or mobilization of Ca^2+^ stored in intracellular stores [90,91].

The activity of the N-methyl-D-aspartate receptors (NMDARs) could also be compromised by pesticide exposure. NMDARs are highly Ca^2+^-permeable channels, widespread in the CNS, where they mediate essential glutamatergic processes such as synaptic transmission and plasticity [92,93,94]. The results obtained by Fortalezas et al. [42] reveal that rotenone induces sustained stimulation of Ca^2+^ influx through NMDARs in cultured cerebellar granule neurons. Since NMDARs and VGCCs are neighboring proteins in the membrane of mature cerebellar granule neurons, it has been postulated that rotenone-induced activation of VGCCs could trigger the release of L-glutamate near NMDARs and thus induce their opening and subsequent Ca^2+^ influx.

Once cytoplasmic Ca^2+^ reaches the threshold level for activating intracellular signals, mechanisms are triggered that promote its return to basal levels. One of these mechanisms is the extrusion of cytosolic Ca^2+^ by Ca^2+^ pumps and exchangers. Ca^2+^-ATPases constitute the main high-affinity Ca^2+^ expulsion system and play an important role in the maintenance of Ca^2+^ homeostasis. There are two families of Ca^2+^-ATPases: plasma membrane Ca^2+^-ATPases (PMCAs) and sarco(endo)plasmic reticulum Ca^2+^-ATPases (SERCAs) [95]. These ionic pumps can remove excess cytosolic Ca^2+^ by either expelling it out of the cytosol into the extracellular milieu or sequestering it in intracellular stores such as the ER [96,97]. Moreover, an important property of PMCA is its stimulation by calmodulin, which increases the enzyme’s affinity for Ca^2+^ and the increase in the maximal transport velocity [98]. However, the functioning of this Ca^2+^ extrusion system could be altered by exposure to certain pesticides. In vivo studies have shown that subcutaneous administration of dichlorvos (6 or 200 mg/kg) or oral administration of carbofuran (1 mg/kg) produced a massive inhibition in PMCA activity in Wistar rats [39,47,50]. The main consequence of this inhibition is the accumulation of Ca^2+^ inside the cell. Similarly, in vitro research by Zaidi et al. [64] found that exposure to low concentrations of paraquat (5–10 μM) increased the basal activity of PMCA two-fold but abolished its sensitivity to calmodulin. However, when paraquat was administered at high concentrations (25–100 μM), it induced inhibition of PMCA and caused its proteolytic degradation.

The Na^+^/Ca^2+^ exchanger (NCX) present in the plasma membrane constitutes another of the basic mechanisms involved in the extrusion of Ca^2+^ from the cellular interior [99]. Under physiological conditions, NCX uses the energy stored in the electrochemical Na^+^ gradient to transport Ca^2+^ into the extracellular medium. However, under certain conditions it can operate in reverse mode and transport Ca^2+^ into the cell [100]. This direction of transport is determined by the relative orientation of the electrochemical gradients of Na^+^ and Ca^2+^ and the membrane potential [101].

NCX could be the target of the toxic action of some pesticides that affect Ca^2+^ homeostasis. An example of this would be ziram, as the in vitro study by Jin et al. [61] showed that this pesticide could inhibit the activity of the NCX3 isoform, both in its Ca^2+^ output mode (direct mode) and in its Ca^2+^ input mode (reverse mode). Likewise, the team of Persson et al. [63] observed that NCX was also one of the targets of rotenone-induced toxicity. The findings of this research also suggest that the reverse mode of NCX operation (of Ca^2+^ influx) might be involved in the neurite degeneration observed after rotenone exposure.

### 3.3. Effects on Intracellular Ca^2+^ Stores

The effects of exposure to different pesticides on intracellular Ca^2+^ reservoirs are shown in Table 3. Some of the organelles present inside the cell, mainly the ER and mitochondria, possess Ca^2+^ uptake mechanisms that allow them to sequester this ion from the cytosol and return its concentrations to resting levels. The ER serves numerous specialized functions in cells, including protein biosynthesis and maintenance of Ca^2+^ homeostasis [102,103,104]. The ER constitutes the main intracellular Ca^2+^ store, and a dysregulation in the Ca^2+^ levels within this organelle has been implicated in the pathophysiology of different neurodegenerative diseases [105,106]. Under physiological conditions, the Ca^2+^ content in the ER is tightly regulated, as alteration of its homeostasis can affect protein folding and trigger a condition called ER stress that can ultimately trigger cell death by apoptosis [107,108]. The ER has been shown to be the predominant Ca^2+^ reservoir in malathion- and TBI-induced Ca^2+^ release in human astrocyte cultures [44,45]. The results of both in vitro studies suggest that the release of Ca^2+^ stored in the ER might be the main mechanism by which these pesticides increase [Ca^2+^]i.

The ER is closely linked to the mitochondria, both by proximity and through Ca^2+^ signaling. Specifically, there are specialized structures called mitochondria-associated membranes, which are responsible for the intimate association between these two organelles and are involved in the control of Ca^2+^ homeostasis [111,112]. Therefore, it is expected that exposure to neurotoxic substances that perturb Ca^2+^ homeostasis and produce ER stress may also negatively affect mitochondrial functioning.

Mitochondria constitute the powerhouse of the cell, as they are responsible for orchestrating the production of most cellular energy in eukaryotic cells [113,114]. However, these organelles also possess a Ca^2+^ buffering capacity that allows them to dynamically manage transient alterations in [Ca^2+^]i and buffer severe overloads of this ion [112].

The kinetic characteristics of Ca^2+^ uptake and release mechanisms in the mitochondria dictate that when [Ca^2+^]i is above the threshold, mitochondria accumulate Ca^2+^ rapidly and release it slowly [115]. This mechanism is in agreement with the mitochondrial behavior observed during [Ca^2+^]i elevations induced by certain pesticides. Specifically, the findings of the in vivo study by Kaur et al. [109] demonstrate that chronic exposure of Wistar rats to dichlorvos (6 mg/kg) significantly increased Ca^2+^ uptake by the mitochondria, while the team of Azevedo et al. [110] found that in the presence of pyriproxyfen (0.01 or 0.1 μg/mL), the mitochondrial Ca^2+^ release was markedly reduced in zebrafish. However, Azevedo et al. [110] also observed that the higher concentration of pyriproxyfen (0.1 μg/mL) reduced the Ca^2+^ uptake into the cell interior, which could be related to a loss of mitochondrial membrane potential, or a damage of the ER caused by the pesticide. These results are in line with the extensive evidence pointing to the mitochondria as one of the targets of neurotoxic action exerted by pesticides [116,117,118,119].

In summary, the studies analyzed in the present review show that pesticide-induced disruption of Ca^2+^ homeostasis could result in dysfunction of the ER and mitochondria, as well as damage to the crosstalk between these two organelles. Specifically, pesticide exposure could induce ER stress and increase Ca^2+^ transfer from the ER lumen to mitochondria, which could result in mitochondrial Ca^2+^ overload [103]. An excess of Ca^2+^ in mitochondria may lead to an impairment of the mitochondrial respiratory chain and increase the generation of ROS that will ultimately lead to cell apoptosis.

### 3.4. Effects on Ca^2+^ Binding Proteins and Intracellular Signaling Pathways

Table 4 summarizes the main results observed on the alterations of the Ca^2+^ binding proteins and intracellular signaling pathways induced by pesticides. Ca^2+^ release from intracellular compartments can be triggered by the activation of some signaling pathways. Among these pathways is phospholipase C (PLC), which plays a critical role in the release of Ca^2+^ stored in the ER. Activation of PLC triggers the formation of IP_3_ which, upon binding to the IP_3_ receptors present on the ER membrane, induces the Ca^2+^ release from this organelle [120,121]. The team of Hsu et al. [45] found that the PLC-dependent Ca^2+^ release was involved in the malathion-induced [Ca^2+^]i increase. These results agree with those obtained in previous studies, in which neuronal damage resulting from organophosphate exposure was associated with an increase in IP_3_ levels [122,123,124].

In the study by Vatanparast et al. [58] paraoxon exposure produced a primary hyperexcitability in snail neurons, followed by a secondary silencing effect due to an increase in the duration of after-potential hyperpolarization. This secondary increase in the duration of afterhyperpolarization was abolished by an antagonist of IP_3_-mediated Ca^2+^ release. These results suggest that paraoxon could elicit slow activation of intracellular cascades that induce IP_3_-mediated Ca^2+^ release from ER stores and consequently increase the duration of afterhyperpolarization.

This phosphoinositol system is well known as a pathway that can be activated by G protein-coupled receptors, including muscarinic acetylcholine receptors [126,127]. Therefore, the interaction of pesticides with muscarinic receptors constitutes one of the mechanisms by which these substances can mobilize Ca^2+^ from ER stores, although this might not be the only way. In a previous study, Katz and Marquis [86] demonstrated that chronic exposure to paraoxon (0.1 nM) increased IP_3_ levels in a time-dependent manner. This effect was partially inhibited by muscarinic antagonists, but completely blocked by a PLC inhibitor, suggesting direct interaction between the pesticide and PLC. Results from other studies also suggest that the paraoxon-enhanced Ca^2+^ release is due to direct interaction between the pesticide and one of the downstream elements of the IP_3_ signaling cascade, presumably the ER receptor [53].

PLC-mediated breakdown of phosphatidylinositol 4,5-bisphosphate not only generates IP_3_, but also produces another messenger, diacylglycerol, which can activate protein kinase C (PKC) [128]. The PKC system possesses the common property of transmitting downstream signals by phosphorylating additional effector proteins or signaling partners [129]. Conventional PKC possess a Ca^2+^-binding domain, so that their activity is regulated by both diacylglycerol and [Ca^2+^]i [130,131]. Some isoforms of PKC possess the ability to modulate Ca^2+^ channels and pumps, as well as families of proteins involved in the maintenance of Ca^2+^ homeostasis [129]. In line with this, Hsu et al. [45] demonstrated that PKC exerts a regulatory role in malathion-induced Ca^2+^ entry through SOCs. Similarly, the team of Vatanparast et al. [58] also observed that PKC modulation modifies Ca^2+^ action potential and neuronal activity but does not contribute to the actions exerted by paraoxon. Therefore, it appears that PKC does not participate in the suppression of the Ca^2+^ action potential induced by paraoxon, although PKC activation could attenuate some of the effects produced by this organophosphate.

On the other hand, the increase in [Ca^2+^]i induced by certain pesticides can lead to the activation of Ca^2+^-activated proteases, mainly calpain. Calpain is an evolutionarily conserved Ca^2+^-dependent cysteine protease that influences a wide range of physiological events, including development, differentiation, cell proliferation, and apoptosis, among others [132,133]. Inappropriate regulation of the proteolytic system involving calpain has been associated with various human pathological disorders, including Alzheimer’s disease and cerebral ischemia [134,135,136].

In line with the above, it has been observed that several organophosphate pesticides were able to significantly increase calpain activity [39,41,50,65]. Calpain activation has been shown to be related to the degradation of cytoskeletal elements and axonal degeneration, a typical feature of organophosphate-induced toxicity [137,138,139,140].

The research findings of Zaidi et al. [64] suggest a dual role of calpain as a function of pesticide concentration. Thus, during exposition to low concentrations of paraquat, calpain appears to cleave the calmodulin binding domain of PMCAs, thereby increasing their activation in order to counteract Ca^2+^ overload and maintain cell viability [141]. However, under severe oxidative stress conditions induced by exposure to higher concentrations of paraquat, calpain appears to cause oxidative modification of PMCAs and their subsequent inactivation [142,143].

Increased [Ca^2+^]i can also influence the activity of Ca^2+^/calmodulin-dependent protein kinases II (CaMKIIs), which are key regulators of Ca^2+^ signaling in eukaryotic cells [144]. This enzyme is activated as a result of increased [Ca^2+^]i and phosphorylates target proteins involved in processes such as ion channel modulation and regulation of gene expression, among others [145,146]. In the study by Liu et al. [49] it was observed that rotenone-induced increase in [Ca^2+^]i leads to CaMKII activation, resulting in inhibition of mammalian or mechanistic target of rapamycin (mTOR) signaling and induction of neuronal apoptosis. Since the protein kinase mTOR acts as a central controller for neuronal survival, suppression of this pathway can trigger apoptosis [147].

S100 calcium-binding protein beta (S100*β*) is a Ca^2+^-binding protein that is expressed at high levels in the brain, where it is synthesized mainly by astrocytes [148]. S100*β* proteins regulate a wide range of intracellular processes such as Ca^2+^ homeostasis, cell growth and differentiation, or transcription, among others [149,150]. However, these proteins are also secreted into extracellular fluids and exert regulatory effects on cellular activity by acting in a paracrine, autocrine, and endocrine manner [151]. Extracellular S100*β* exerts a dual effect on neurons depending on its concentration, being neurotrophic at physiological levels (nanomolar) and neurotoxic when at higher concentrations (micromolar) [152]. For this reason, elevated levels of S100*β* in biological fluids is considered a biomarker of neuronal damage [153,154].

Deltamethrin exposure has been shown to cause positive up-regulation of S100*β* expression in both radial glial cells and astrocytes of Wistar rats on postnatal days 12 and 15 [125]. This positive regulation was most prominent in pesticide-treated animals during the second postnatal week. It has been evidenced that S100*β* stimulates increases in intracellular free Ca^2+^ levels in both glial cells and neurons [155]. Therefore, pesticide-induced up-regulation of S100*β* could further potentiate the increase in [Ca^2+^]i and consequent neurotoxicity in early developmental stages.

### 3.5. Protective Effects of Some Treatments on Pesticide-Induced Alterations in Ca^2+^ Homeostasis

The beneficial effects of treatment with Ca^2+^ chelators and antioxidant substances on the pesticide-induced neuronal toxicity are shown in Table 5. The set of results previously presented suggests that most pesticides exert part of their neurotoxic action by elevating [Ca^2+^]i. Consequently, one of the strategies that showed the greatest efficacy in reversing pesticide-induced alterations has been the use of Ca^2+^ chelators. Such chelators mainly include 1,2-bis-(2-aminophenoxy)ethane-N,N,N′,N′-tetraacetic acid acetoxymethyl ester (BAPTA-AM) and ethylene glycol-bis(*β*-aminoethyl ether)-N,N,N′,N′-tetraacetic acid (EGTA). Treatment with BAPTA-AM, a selective intracellular Ca^2+^ chelator, showed protective potential to prevent cytotoxicity evoked by malathion, rotenone and lambda-cyhalothrin [44,45,49]. Likewise, the team of Liu et al. [49] also found that prevention of extracellular Ca^2+^ entry by treatment with EGTA (an extracellular Ca^2+^ chelator) significantly attenuated rotenone-induced cell death. Taken together, these results suggest that the cytotoxicity induced by these pesticides may be closely related to increases in [Ca^2+^]i. In contrast, the study by Vatanparast et al. [59] has shown that paraoxon continued to decrease the duration and amplitude of Ca^2+^ spikes even after BAPTA administration, suggesting that this pesticide may have a profound effect on neuronal activity by blocking VGCCs.

On the other hand, in the research of Kamboj and Sandhir [47], it has been shown that N-acetylcysteine treatment had a beneficial effect on carbofuran-induced alterations in Ca^2+^ homeostasis. N-acetylcysteine is a precursor of the natural antioxidant glutathione that can neutralize free radicals before they can cause cell damage [156,157]. It has been reported that oxidative stress can increase [Ca^2+^]i by inducing its release from intracellular stores and/or its entry from the extracellular medium [158]. There is ample evidence that pesticides can significantly increase free radical levels in cells, so these oxidative stress conditions could favor alterations in Ca^2+^ homeostasis [159]. Thus, the antioxidant properties of N-acetylcysteine could confer it a beneficial potential against the alteration of Ca^2+^ homeostasis, as already demonstrated in previous studies [160,161].

Similarly, Li et al. [48] demonstrated that pretreatment with tert-butylhydroquinone significantly reduced the increase in [Ca^2+^]i and ROS generation induced by deltamethrin. Tert-butylhydroquinone is a food additive known for its antioxidant properties [162] and for its ability to activate the nuclear factor erythroid 2-related factor 2 (Nrf2)-mediated transcription, one of the key factors in preventing excessive oxidative stress in brain cells [163,164,165]. Likewise, Nrf2 has also been shown to play a relevant role in the maintenance of Ca^2+^ homeostasis, as its inactivation/depletion has been linked to significant increases in [Ca^2+^]i [166,167]. Therefore, administration of Nrf2 agonists, such as tert-butylhydroquinone, could constitute an effective approach for the restoration of pesticide-induced alterations in Ca^2+^ homeostasis.

The team of Fortalezas et al. [42] observed that creatine treatment was able to markedly attenuate the early [Ca^2+^]i dysregulation induced by acute rotenone exposure. Creatine is an endogenous compound that is predominantly stored in skeletal muscles but is also present in other parts of the body, such as the liver, kidneys, or brain [168,169]. The predominant role of creatine is to maintain energy homeostasis by keeping ATP levels constant in cells with high energy demands, such as neurons [170]. In addition, this endogenous compound can also reduce free radical formation and promote the restoration of Ca^2+^ homeostasis by decreasing both intracellular and mitochondrial Ca^2+^ levels [168,171]. Therefore, creatine’s ability to promote mitochondrial functioning, attenuate oxidative stress, and restore Ca^2+^ homeostasis makes it a potential candidate for the treatment of pesticide-induced neuropathological alterations.

## 4. Discussion

The data described in all the studies included in this systematic review coincide in pointing out the ability of different classes of pesticides to induce alterations in the Ca^2+^ homeostasis. The pesticide-induced alterations in the [Ca^2+^] in neurons documented in the in vivo studies provide sufficient evidence about the ability of these substances to cross the blood–brain barrier and enter the CNS, where they can produce significant neurotoxic effects.

Ca^2+^ acts as a second messenger in many neuronal signaling pathways, where it directly or indirectly influences many cellular functions. For this reason, cells possess multiple mechanisms aimed at maintaining Ca^2+^ homeostasis, since an alteration in its levels can have significant harmful effects. During nervous system development, Ca^2+^ is responsible for orchestrating a wide range of biochemical cascades and gene expression programs critical for neurodevelopment [172,173]. Therefore, an alteration of Ca^2+^ levels during this stage due to pesticide exposure could cause significant alterations in the developmental process and favor the onset of various disorders. In this sense, the lower capacity to metabolize pesticides during early age could increase vulnerability to the toxic effects of these substances. This has been observed in the study by Meijer et al. [54], where parental compounds of some pesticides seem to disrupt Ca^2+^ levels to a greater extent compared to their metabolites.

Under physiological conditions, cells maintain a basal [Ca^2+^]i of approximately 100 nM, by storing it in intracellular organelles and expelling it into the extracellular space [174]. The studies reviewed here suggest that pesticide exposure can alter the [Ca^2+^]i by perturbing the mechanisms involved in the Ca^2+^ homeostasis and increasing its intracellular levels. Figure 4 shows the main biochemical mechanisms by which pesticides produce these effects on neuronal Ca^2+^ homeostasis.

As previously mentioned, an increase in the [Ca^2+^]i can occur due to the influx of the ion from the extracellular medium through plasma membrane channels and/or due to a Ca^2+^ release from internal stores. Ca^2+^ influx from the extracellular space may occur mainly through VGCCs or ROCs. It has been demonstrated that some organophosphate pesticides can activate VGCCs, especially L- and T-type VGCCs, thereby enhancing Ca^2+^ influx into the cellular interior. There is also evidence about the influence of this class of pesticides in the activation of nAChRs and mAChRs, whose activity is related to the Ca^2+^ influx or its release from intracellular stores, respectively. However, the action of organophosphates on these receptors appears to vary, and they may act as agonists or inhibitors depending on the particular organophosphate, receptor subtype, and cell type. Pesticides also appear to induce the opening of other channels present in the plasma membrane, such as TRPs, non-selective cation channels that allow Ca^2+^ influx into certain cell types.

When an increase in the [Ca^2+^]i levels occurs, various types of proteins are activated to expel the ion against the gradient by means of ATP hydrolysis (PMCAs) or by using the energy stored in the electrochemical Na^+^ gradient (NCXs) [96,100]. The finding reviewed here points to the ability of pesticides to decrease the activity of PMCAs, involved in Ca^2+^ extrusion. In addition, some of these compounds also altered the mode of operation of NCX, either by inhibiting its two modes of action (Ca^2+^ export and import) or by reversing its normal mode of action and causing Ca^2+^ import. While under physiological conditions, high [Ca^2+^]i are rapidly eliminated to return to basal levels, an alteration in Ca^2+^ extrusion systems by pesticides will increase the permanence of the ion inside the cell.

On the other hand, Ca^2+^ release from intracellular stores represents the second basic mechanism to increase its concentration in the cytosol. ER constitutes the main intracellular store of Ca^2+^ and the SERCAs are responsible for the Ca^2+^ accumulation into this organelle [106]. Furthermore, the ER also expresses relatively high amounts of Ca^2+^-binding proteins that enable the storage of Ca^2+^ in its lumen [105]. The membrane of the ER expresses two types of channels that allow Ca^2+^ release into the cytosol: IP_3_ receptors and ryanodine receptors [175]. The evidence reviewed suggests that the ER may be the dominant reservoir in the Ca^2+^ release induced by some pesticides such as malathion, lambda-cyhalothrin, dichlorvos, and pyriproxyfen. Specifically, these pesticides seem to decrease the ability of the ER to sequester Ca^2+^ through inhibition of SERCAs and, at the same time, induce the release of Ca^2+^ from ER through stimulation of IP_3_ and ryanodine receptors.

One of the main pathways involved in Ca^2+^ release from the ER is through PLC activation. PLC activation increases the levels of diacylglycerol and IP_3_, which can trigger different cellular events. IP_3_ can bind to their receptors at the ER membrane, causing its opening and the release of Ca^2+^ into the cytosol. This PLC-dependent pathway appears to be one of those involved in pesticide-induced [Ca^2+^]i increases. However, it is possible that pesticides may induce the activation of this pathway by different ways: (1) by activating G-protein-coupled receptors on the plasma membrane (mainly muscarinic receptors); (2) by directly interacting with PLC; (3) by directly interacting with IP_3_ receptors.

Alterations in the ER Ca^2+^ stores induced by pesticides can lead to the depletion of the ion. To protect itself from the deleterious effects of Ca^2+^ store depletion, the cell has a store-operated gating mechanism, the SOCs, present in the plasma membrane [76]. So, when the content of intracellular Ca^2+^ begins to decline, the SOCs are activated and the Ca^2+^ influx increases [77,176]. Several studies have shown that increases in [Ca^2+^]i induced by some pesticides were mediated by ion entry through the SOCs. Furthermore, Ca^2+^ influx through these channels could be also regulated by PKCs, a large family of proteins that plays an important role in modulating the activity of direct Ca^2+^ regulators such as Ca^2+^ channels [129].

In addition to the ER, mitochondria also play a key role in cytosolic Ca^2+^ uptake [112]. Within the mitochondria, Ca^2+^ plays a key role regulating the activity of enzymes operating in the respiratory chain and that are essential for ATP production [177,178]. However, when cytoplasmic Ca^2+^ levels rise above the threshold, mitochondria can accumulate Ca^2+^ rapidly and release it slowly [115]. This alteration in mitochondrial dynamics has been observed following exposure to pesticides, which can lead to significant Ca^2+^ overload within these organelles. Mitochondrial Ca^2+^ overload is severely detrimental to mitochondrial function and leads to a significant increase in ROS production [179]. This increase in mitochondrial ROS can potentiate the increase in Ca^2+^ levels, creating a self-amplifying feedback loop that causes further cellular damage [180]. Finally, increased ROS and Ca^2+^ loading can trigger the opening of the mitochondrial permeability transition pore, resulting in the release of proapoptotic factors into the cytoplasm [181].

In line with the above, it has been widely demonstrated that pesticide exposure favors ROS production and, with it, oxidative stress conditions [159,182,183]. ROS can modulate the activity of a wide variety of Ca^2+^ channels, pumps, and exchangers. There is evidence that ROS can stimulate Ca^2+^ entry through some VGCCs, TRPs and SOCs [181]. Likewise, ROS can also inhibit the activities of SERCAs and PMCAs, compromising the ER Ca^2+^ sequestration and favoring the permanence of the ion in the intracellular milieu, respectively [158,184]. Therefore, pesticide-induced oxidative stress conditions can potentiate both the rise and permanence of increased Ca^2+^ levels.

Increased [Ca^2+^]i can also trigger the activation of different proteins such as calpain, a Ca^2+^-activated cysteine protease involved in a wide range of physiological events [132,133]. Under conditions of moderate oxidative stress, calpain activation could exert a defensive function by enhancing the activation of PMCAs to remove excess of Ca^2+^ from cytosol [64]. However, when oxidative stress is prolonged and/or severe, calpains can trigger the degradation of PMCAs and cellular cytoarchitectural elements, ultimately leading to apoptosis [39,41,64,185].

Increased intracellular free Ca^2+^ can also activate of CAMKII, an important signal transducer molecule that phosphorylates channels and targets proteins involved in various physiological processes [145,146]. So, activated CAMKII can phosphorylate specific subunits of NMDARs and VGCCs to promote Ca^2+^ influx and thereby enhance the increase in [Ca^2+^]i [186,187,188]. Likewise, another molecular target of CAMKII is the mTOR protein, which is involved in the regulation of signaling pathways essential for neuronal survival [147]. The evidence reviewed suggests that exposure to certain pesticides can induce CAMKII activation, which could exert an inhibitory effect on mTOR and thus favor apoptosis [49].

Increased expression of the S100*β* protein was another change caused by pesticides. This multifunctional protein could exert a dual neurotoxic effect on neurons. On the one hand, extracellular S100*β* could increase ROS levels inside neurons by binding to the membrane receptors for advanced glycation end products (RAGEs) [151], while the S100*β* protein found inside neurons could potentiate the increase in [Ca^2+^]i [155].

However, the increase in [Ca^2+^]i was not observed in all of the studies analyzed, with organophosphates being the class of pesticides that exerted the most imprecise effects on the Ca^2+^ levels. Therefore, although most of the evidence included here suggests that pesticides induce an increase in Ca^2+^ levels, further research is needed to confirm whether the differences found between studies could be due to differences between species, pesticide class, dose and/or exposure characteristics used in each study.

Finally, intracellular Ca^2+^ overload and excessive ROS generation seem to be the two key elements of neuropathology caused by pesticide exposure. For this reason, the Ca^2+^ chelators (intra- and extracellular) and molecules with antioxidant properties represent two potential therapeutic candidates, which have already been shown to be effective in reducing neurotoxicity caused by pesticides.

## 5. Limitations

The main limitation observed in the present systematic review is that the findings on the effects of pesticides have been obtained through studies carried out in a wide variety of species, both vertebrates and invertebrates. Therefore, various methodological aspects must be considered before extrapolating the results obtained to human problems. The main problems in generalizing the results lie in the clear biological differences between the species, as well as the doses of pesticide and the routes of administration used.

In line with the above, in the small number of in vivo studies carried out with rodents, the subcutaneous and intraperitoneal routes of administration were used, which are not representative of the usual modes of exposure in humans. Only in one study on rodents was the pesticide administered orally, which could be useful to assess the neurotoxic consequences of consuming food contaminated with pesticides. However, there are other routes of exposure that have not been investigated, such as the respiratory route.

On the other hand, most of the studies with mammals were carried out with cell cultures, which reduces the possibility of extrapolating the findings to the real effect produced by exposure to these substances in humans. Furthermore, some of the in vivo studies describe the effects of exposure to higher doses than those to which the general population is routinely exposed. Therefore, it is possible that the severity of the neurotoxic effects studied may vary depending on the dose.

## 6. Conclusions

The findings described in the present review provide ample evidence for the ability of different classes of pesticides to alter [Ca^2+^]i homeostasis in neurons. Most of the pesticides analyzed seem to induce a significant increase in [Ca^2+^]i by activating membrane Ca^2+^ channels and inhibiting Ca^2+^ extrusion mechanisms; at the same time, they favor the Ca^2+^ release from intracellular stores, especially from the ER. An excessive decrease in Ca^2+^ stored in the ER can induce the response to stress in this organelle and prompt the mitochondria to assume the storage of large amounts of Ca^2+^ coming from both the ER and the cytosol. Ca^2+^ overload in mitochondria would induce a significant impairment in the functioning of the mitochondrial respiratory chain, severely disturbing ATP synthesis and increasing ROS generation. Generally, under these oxidative stress conditions, oxidants activate Ca^2+^ channels, inhibit pumps, and can reverse Na^+^/Ca^2+^ exchangers, thus potentiating the increase in [Ca^2+^]i. Likewise, ROS cause deterioration of membrane lipids, favoring Ca^2+^ influx from the extracellular medium, in addition to damaging the membranes of organelles that stores Ca^2+^, potentiating the leakage of the ion into the cytosol. As a result of oxidative stress and intracellular Ca^2+^ overload in the mitochondria and nucleus, a cascade of biochemical processes is activated, including calpain activation and CAMKII-dependent mTOR inhibition, leading ultimately to neuronal death by apoptosis.

However, some pesticides did not increase the [Ca^2+^]i or even showed contradictory results, especially organophosphate compounds. Therefore, the discrepancy between some findings points to the need for further studies to identify how factors such as the type of pesticide, the dose, or the mode of exposure can modify the effects of these compounds on Ca^2+^ homeostasis in the CNS.

## Figures and Tables

**Figure 1 ijms-22-13376-f001:**
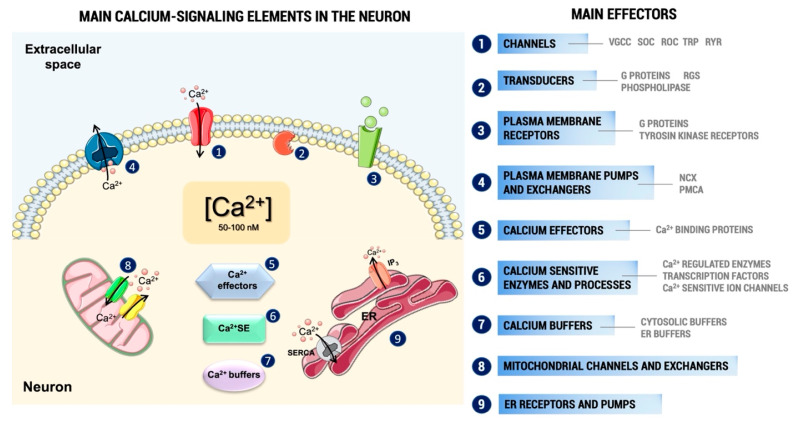
Increases in cytosolic Ca^2+^ levels can take place through activation of various types of channels and receptors present in the plasma membrane, including VGCC, ROC, SOC, or transducers. Regarding the release of Ca^2+^ from internal compartments, this ion can come mainly from the ER or the mitochondria through activation of ER receptors and pumps or mitochondrial channels and exchangers. Most of this Ca^2+^ is bound to buffers, whereas a small proportion binds to the effectors that activate various cellular processes that operate over a wide temporal spectrum. In the process of decreasing its intracellular levels, Ca^2+^ leaves the effectors and buffers and is eliminated from the cell by various plasma membrane exchangers and pumps as NCX and PMCA, which extrude Ca^2+^ to the outside the neuron. Other mechanisms used to decrease cytosolic Ca^2+^ levels are its accumulation in organelles such as ER, through SERCA, and mitochondrial pumps and exchangers. Parts of the figure were created using templates from Servier Medical Art, which are licensed under a Creative Com-mons Attribution 3.0 Unported License (http://smart.servier.com/, accessed on 4 December 2021). **Abbreviations:** VGCC, voltage-gated calcium channel; ROC, receptor operated channel; SOC, store-operated cannel; ER, endoplasmic reticulum; NCX, Na^+^/Ca^2+^ exchanger; PMCA, plasma membrane Ca^2+^-ATPase; SERCA, sarco(endo)plasmic reticulum Ca^2+^-ATPase; Ca^2+^ SE, Ca^2+^ sensitive enzymes; TRP, transient receptor potential channels; RYR, ryanodine receptors; RGS, regulators of G-protein signaling.

**Figure 2 ijms-22-13376-f002:**
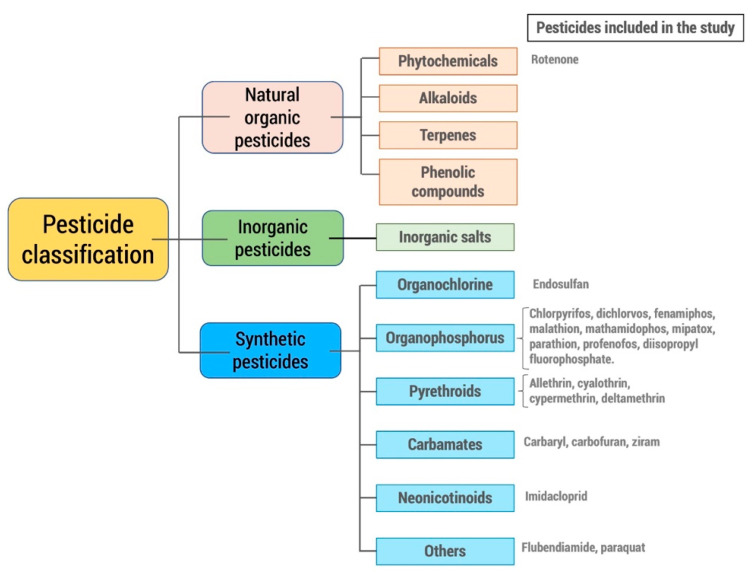
Classification of pesticides according to their origin and chemical group; pesticides included in the review.

**Figure 3 ijms-22-13376-f003:**
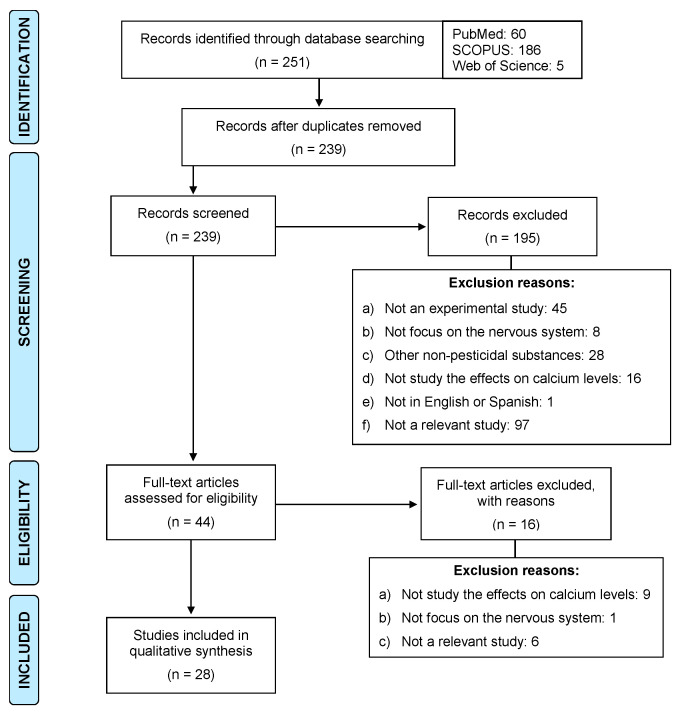
Flow diagram of the systematic search process.

**Figure 4 ijms-22-13376-f004:**
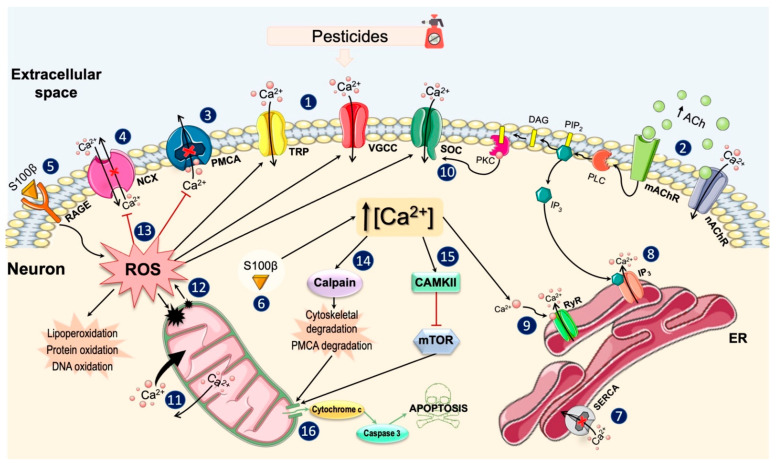
Main mechanisms of action of pesticides on neuronal Ca^2+^ homeostasis. Exposition to pesticides induces a series of changes in the plasma membrane that include: (**1**) opening of the VGCCs (especially the L- and T- types) and some TRP channels, which allow the Ca^2+^ influx and enhance the membrane depolarization; (**2**) activation of nAChRs and mAChRs, by increasing the availability of ACh and/or by binding directly to these receptors; (**3**) inhibition of PMCA, the main Ca^2+^ extrusion mechanism; (**4**) alteration of the NCX, completely inhibiting its activity or activating its reverse mode. Pesticides also increase the S100*β* levels which: (**5**) binds to RAGE in the extracellular side and favors the production of ROS and (**6**) in the intracellular medium, it increases the Ca^2+^ levels. In the cytosol, pesticides induce the depletion of ER Ca^2+^ reserves by: (**7**) inhibiting the SERCAs, responsible for sequestering Ca^2+^; (**8**) stimulating the IP_3_-induced Ca^2+^ release (**9**); and Ca^2+^ release through RyRs stimulated by cytosolic Ca^2+^. When the Ca^2+^ content of the ER begins to decline, the PKCs stimulate the influx of Ca^2+^ through the SOCs (**10**) and the mitochondria assume the role of Ca^2+^ reservoir, rapidly accumulating large amounts of Ca^2+^ and slowly releasing it (**11**). The overload of Ca^2+^ in the mitochondria increases ROS levels and its release to the cytosol (**12**), where they enhance the [Ca^2+^]i by stimulating Ca^2+^ channels and inhibiting their expulsion mechanisms, damaging lipids, cell proteins, and DNA (**13**). The increases in ROS and Ca^2+^ levels activate calpains and CAMKII. Calpains induce the degradation of elements of the cellular cytoarchitecture (**14**), while CAMKII inhibits mTOR (**15**). These two pathways can ultimately cause the release of pro-apoptotic factors from the mitochondria, finally leading to cell death (**16**). Parts of the figure were created using templates from Servier Medical Art, which are licensed under a Creative Commons Attribution 3.0 Unported License (http://smart.servier.com/, accessed on 29 October 2021). **Abbreviations**: VGCC, voltage-gated calcium channel; TRP, transient receptor potential; nAChR, nicotinic acetylcholine receptor; mAChR, muscarinic acetylcholine receptor; ACh, acetylcholine; PMCA, plasma membrane Ca^2+^-ATPase; NCX, Na^+^/Ca^2+^ exchanger; S100*β*, S100 calcium-binding protein beta; RAGE, receptor for advanced glycation end products; ROS, reactive oxygen species; ER, endoplasmic reticulum; SERCA, sarco(endo)plasmic reticulum Ca^2+^-ATPase; IP_3_, inositol 1,4,5-trisphosphate; RyR, ryanodine receptors; PKC, protein kinase C; SOC, store-operated channel; CAMKII, Ca^2+^/calmodulin-dependent protein kinase II; mTOR, mammalian or mechanistic target of rapamycin; PLC, phospholipase C; PIP_2_, phosphatidylinositol-4,5-bisphosphate; DAG, diacylglycerol.

**Table 1 ijms-22-13376-t001:** General changes induced by pesticides in the Ca^2+^ homeostasis.

Species or Cellular Line	Dose and Time of Exposure	Objective	Results	Reference
Wistar rat	Dichlorvos: 6 mg/kg s.c. for 8 weeks	To investigate alterations in neuronal Ca^2+^ homeostasis	↑ [Ca^2+^]i in brain stem and cerebellum	[50]
Wistar rat	Dichlorvos: 200 mg/kg s.c. in a single dose	To examine the role of the Ca^2+^ messenger system in the development of delayed neurotoxicity	↑ [Ca^2+^]i	[39]
Wistar rat	Carbofuran: 1 mg/kg oral for 28 days	To study the alterations in Ca^2+^ homeostasis and neurobehavioral deficits induced by the pesticide	↑ [Ca^2+^]i in the synaptosomes	[47]
Rat CGN	Rotenone: 2–50 nM for 30 min or 12 h	To assess the effects of rotenone at very low concentrations in mature CGN	↑ [Ca^2+^]i induced by rotenone (10 and 15 nM) at 30 min	[42]
Rat midbrain slices	Rotenone: 0.05–1 μM for 10 min	Investigate the effects of rotenone on individual neurons of the rat substantia nigra *pars compacta*	↑ [Ca^2+^]i	[43]
Mouse DRG	Malathion: 0.1–100 μM for 0–16 min	To assess the role of TRPA1 in organophosphate-induced delayed neuropathy	↑ [Ca^2+^]i and upregulation of neuronal excitability	[40]
Mouse PC12 cells and primary neurons	Rotenone: 0–1 μM for 24 h	To investigate whether rotenone induces apoptosis by inhibition of the Ca^2+^/ROS-dependent mTOR pathway	↑ [Ca^2+^]i ↑mitochondrial H_2_O_2_ levels, which induced ↑ [Ca^2+^]i	[49]
PC12 cells	Carbaryl, chlorpyrifos, parathion-ethyl and its metabolites–oxon: 0.1–10 µM for 20 min	To evaluate the effects of several pesticides and their metabolites on the basal [Ca^2+^]i	- All OPs inhibited the depolarization-induced ↑ [Ca^2+^]i - The parent compounds were more potent than their –oxon metabolites in altering the [Ca^2+^]i - The mixtures of chlorpyrifos+oxon analog or +parathion did not increase the degree of inhibition on the ↑ [Ca^2+^]i	[54]
PC12 cells	DLT: 10 µM for 1 h	To investigate the neuroprotective effect of tert-butylhydroquinone against oxidative stress induced by DLT	↑ [Ca^2+^]i	[48]
GHA and human glioblastoma DBTRG-05MG cells, and D1 TNC1 rat astrocytes	Malathion: 5–25 µM	To explore the mechanism underlying the effects of malathion on Ca^2+^ homeostasis and cell viability	Concentration-dependent ↑ [Ca^2+^]i in GHA cells	[45]
GHA and D1 TNC1 cells	LCT: 10–15 µM	To explore whether LCT affects Ca^2+^ homeostasis and cell viability	↑ [Ca^2+^]i	[44]
SH-SY5Y	Mipaxon, paraoxon: 0.05–2 mM for 4 days	To characterize the cellular targets of organophosphate neurotoxicity	- Paraoxon induced a transient ↑ [Ca^2+^]i - Repeated treatment with paraoxon (0.05 mM) ↓ [Ca^2+^]i in the NGF-differentiated cells	[46]
SH-SY5Y	Mipafox, paraoxon, fenamiphos, profenofos: 1 × 10^−10^–1 × 10^−2^ M for 24 or 48 h	To evaluate the neurotoxic effects of mipafox and paraoxon, as well as the potential of fenamiphos and profenofos to cause acute and/or delayed effects	Both mipafox and fenamiphos ↑ [Ca^2+^]i	[41]
SH-SY5Y and CCF-STTG1	Paraoxon, DFP: 0.3, 1, 3, 10 or 30 μM for 1–4 days	To compare the neurotoxic effects of paraoxon and DFP in two cell lines	Paraoxon (1–30 µM), but not DFP, ↓ the mitochondrial:cytosolic Ca^2+^ ratio in TLC cultures	[55]
Domestic honeybees (*Apis mellifera*)	Flubendiamide: 3 µM	To evaluate the effects of the insecticide on normal Ca^2+^ homeostasis in antennal neurons of honeybees	Strong Ca^2+^ transients in antennal neurons	[56]
*Apis mellifera ligustica Spinola*	DLT: 0–250 mg/L for 200 seg or 5 min	To investigate the effect of DLT on the Ca^2+^ channel in nerve cells of the brain	↑ [Ca^2+^]i even with the lowest pesticide concentrations	[37]
*Drosophila melanogaster*	Ziram: 20 μM	To compare the effects of ziram on type II aminergic versus type Ib glutamatergic nerve endings	Spontaneous and synchronized bursts of Ca^2+^ input and electrical activity in type II, but not in type Ib terminals	[57]
Snail neurons	Paraoxon: 0.3–0.6 µM for 10 min	To investigate the interaction of paraoxon with PKC and the release of Ca^2+^ mediated by IP_3_, on the modulation of action potentials and neuronal activity	↓ Duration of Ca^2+^ action potentials and ↓ duration of PHP, associated with an ↑ in firing frequency Paraoxon (0.6 μM) ↓ the duration of action potentials, but ↑ the duration of PHP, along with a ↓ in the firing rate	[58]
Neuronal soma of land snail (*Caucasotachea atrolabiata*)	Paraoxon: 0.3 µM for 5 or 10 min	To study the effects of the pesticide on Ca^2+^ spikes and neuronal excitability in snail neurons	Paraoxon (0.3 μM) reversibly ↓ the duration and amplitude of the Ca^2+^ peaks ↓ in the duration and amplitude of PHP, leading to a significant ↑ in the frequency of Ca^2+^ peaks	[59]

**Abbreviations:** CGN, cerebellar granule neurons; DRG, dorsal root ganglion; TRPA1, transient receptor potential cation channel subfamily A member 1; mTOR, mammalian target of rapamycin; OP, organophosphate pesticide; ROS, reactive oxygen species; DLT, deltamethrin; GHA, Gibco^®^ human astrocytes; LCT, lambda-cyhalothrin; NGF, nerve growth factor; DFP, diisopropyl fluorophosphate; PKC, protein kinase C; IP_3_, inositol 1,4,5-trisphosphate; PHP, posthyperpolarization.

**Table 2 ijms-22-13376-t002:** Alterations in the functioning of membrane channels and pumps.

Species or Cellular Line	Dose and Time of Exposure	Results	Reference
Wistar rat	Dichlorvos: 6 mg/kg s.c. for 8 weeks	↑ Ca^2+^ influx through the VGCC ↓ Ca^2+^-ATPase activity	[50]
Wistar rat	Dichlorvos: 200 mg/kg s.c. in a single dose	↓ Ca^2+^-ATPase activity	[39]
Sprague Dawley rat	Allethrin, cyhalothrin, DLT: 10, 20 or 60 mg/kg i.p. in a single dose	Nimodipine completely blocked the glutamate release induced by DLT (60 mg/kg)	[60]
Wistar rat	Carbofuran: 1 mg/kg oral for 28 days	↓ Ca^2+^-ATPase activity with a concomitant ↑ in K^+^-induced Ca^2+^ influx through VGCCs	[47]
Mouse primary ventral midbrain neurons	Ziram: 10 mM	Dopaminergic neurons lacking NCX3 were less sensitive to ziram-induced neurotoxicity	[61]
PC12 cells	Carbaryl, chlorpyrifos, parathion-ethyl and its metabolites –oxon: 0.1–10 μM for 20 min	- The parent compounds were more potent than their –oxon metabolites in altering the [Ca^2+^]i - The mixtures of chlorpyrifos+oxon analog or +parathion did not increase the degree of inhibition on the ↑ [Ca^2+^]i	[54]
PC12 cells and rat primary cortical cells	Endosulfan, cypermethrin, chlorpyrifos, chlorpyrifos-oxon, carbaryl, and IMI: 0.1–100 μM for 24 h (and 20 min in the second exposure)	- All insecticides (except carbaryl and IMI) induced slow or non-reversible VGCCs inhibition (subchronic conditions) - Chlorpyrifos was clearly more potent in inhibiting VGCCs in the repeated exposure compared to acute exposure	[62]
Mouse DRG	Rotenone: 1 μM for 3 or 6 days	- NCX reverse mode inhibition protected against rotenone-exposed neurites from degeneration - Rotenone exposure was associated with delayed Ca^2+^ elimination after neurite activation	[63]
Rat CGN cells	Rotenone: 2–50 nM for 30 min or 12 h	Nifedipine and, to a lesser extent, MK-801 attenuated the rotenone-induced alteration in the Ca^2+^ homeostasis	[42]
Rat midbrain slices	Rotenone: 0.05–1 μM for 10 min	The rotenone-induced ↑ [Ca^2+^]i was blocked by eliminating extracellular Ca^2+^ and was attenuated by a TRPM2 blocker	[43]
Rat primary cortical neurons	Paraquat: 5–100 μM for 5 min, 15 min or 24 h	- Paraquat (5–10 μM) doubled the basal activity of PMCA, but abolished its sensitivity to calmodulin - Paraquat (25–100 μM) ↓ PMCA activity and were associated with the formation of high molecular weight PMCA aggregates	[64]
Mouse DRG	Malathion: 0.1–100 μM for 0–16 min	Malathion-induced Ca^2+^ influx currents were attenuated by a TRPA1 antagonist and eliminated by suppression of *Trpa1.gene*	[40]
Hens (*Leghorn isabrown*)	Methamidophos: 50 mg/kg oral for 1 or 21 days	Nimodipine ↓ alterations induced by the pesticide	[65]
GHA and human glioblastoma DBTRG-05MG cells, and D1 TNC1 rat astrocytes	Malathion: 5–25 μM	- The malathion-induced ↑ [Ca^2+^]i was reduced by eliminating the Ca^2+^ from extracellular medium - Malathion-induced ↑ [Ca^2+^]i was inhibited by blockers of SOCs	[45]
GHA and D1 TNC1 cells	LCT: 10–15 μM	LCT-induced ↑ [Ca^2+^]i was reduced by eliminating extracellular Ca^2+^ and was inhibited by modulators of SOCs	[44]
SH-SY5Y	Mipaxon, Paraoxon: 0.05–2 mM for 4 days	Paraoxon (0.05 mM) attenuated the transient ↑ [Ca^2+^]i induced by carbachol	[46]
*Apis mellifera ligustica Spinola*	DLT: 0–250 mg/L for 200 seg or 5 min	DLT had toxic effects on T-type VGCCs, but not on L-type VGCCs, channels activated by NMDAR or Ca^2+^ store	[37]
Neuronal soma of land snail (*Caucasotachea atrolabiata*)	Paraoxon: 0.3 μM for 5 or 10 min	- Apamine ↓ the duration and amplitude of PHP and ↑ the frequency of the peaks - In the presence of apamine, paraoxon ↓ the duration of the Ca^2+^ peak and PHP and ↑ the frequency of neuronal activation	[59]

**Abbreviations:** VGCC, voltage-gated calcium channel; DLT, deltamethrin; NCX3, isoform 3 of the Na^+^-Ca^2+^ exchanger; IMI; imidacloprid; DRG, dorsal root ganglion; CGN, cerebellar granule neurons; NMDAR, N-methyl-D-aspartate receptor; TRPM2, transient receptor potential cation channel subfamily M member 2; PMCA, plasma membrane Ca^2+^-ATPase; DRG, dorsal root ganglion; TRPA1, transient receptor potential cation channel subfamily A member 1; GHA, Gibco^®^ human astrocytes; LCT, lambda-cyhalothrin; SOC, store-operated channel; SK, small-conductance calcium-activated potassium channels; PHP, posthyperpolarization.

**Table 3 ijms-22-13376-t003:** Changes in the intracellular Ca^2+^ stores.

Species or Cellular Line	Dose and Time of Exposure	Results	Reference
Wistar rat	Dichlorvos: 6 mg/kg s.c. for 12 weeks	↑ influx of Ca^2+^ to mitochondria	[109]
GHA and human glioblastoma DBTRG-05MG cells, and D1 TNC1 rat astrocytes	Malathion: 5–25 μM	In a Ca^2+^-free medium, the pretreatment with tapsigargin, a SERCA inhibitor, abolished the pesticide-induced ↑ [Ca^2+^]i Incubation with malathion abolished the tapsigargin-induced ↑ [Ca^2+^]i	[45]
GHA and D1 TNC1 cells	LCT: 10–15 μM	In a Ca^2+^-free medium the pretreatment with tapsigargin suppressed LCT-induced ↑ [Ca^2+^]i Incubation with LCT abolished the tapsigargin-induced ↑ [Ca^2+^]i	[44]
Zebrafish (*Danio rerio*)	Pyriproxyfen: 0.001–10 μmol/L for 1 h (in vitro essay) 0.001, 0.01 or 0.1 μg/mL for 16 h (in vivo essay)	Pyriproxyfen (0.1 μg/mL) ↓ Ca^2+^ uptake by up to 50% Pyridoxyphene (0.01 or 0.1 μg/mL) ↓ mitochondrial Ca^2+^ release by approximately 80%	[110]

**Abbreviations:** GHA, Gibco^®^ human astrocytes; SERCA, sarco(endo)plasmic reticulum Ca^2+^-ATPase; LCT, lambda-cyhalothrin; IP_3_, inositol 1,4,5-trisphosphate.

**Table 4 ijms-22-13376-t004:** Changes in Ca^2+^-binding proteins or other intracellular proteins.

Species or Cellular Line	Dose and Time of Exposure	Results	Reference
Wistar rat	Dichlorvos: 6 mg/kg s.c. for 8 weeks	↑ calpain activity	[50]
Wistar rat	Dichlorvos: 200 mg/kg s.c. Single dose	↑ calpain activity	[39]
Wistar rat	DLT: 0.7 mg/kg i.p. from PND0 until PND7 (DLT-I) or from PND9 until PND13 (DLT-II)	DLT ↑ expression of S100*β* in radial glial fibers and in astrocytes on PND12 and PND15 days The up-regulation of S100*β* was more prominent in the DLT-II group	[125]
Rat primary cortical neurons	Paraquat: 5–100 μM for 5 min, 15 min or 24 h	Proteolytic degradation of PMCA was prevented by a calpain inhibitor	[64]
PC12 cells and rat primary cortical cells	Rotenone: 0–1 μM for 24 h	Rotenone-induced ↑ [Ca^2+^]i activated CaMKII and caused inhibition of mTOR signaling	[49]
Hens (*Leghorn isabrown*)	Methamidophos: 50 mg/kg oral for 1 or 21 days	The (+) and (-) isoforms of methamidophos produced a slight ↑ in calpain activity	[65]
GHA and human glioblastoma DBTRG-05MG cells, and D1 TNC1 rat astrocytes	Malathion: 5–25 μM	Inhibition of PLC blocked the ↑ [Ca^2+^]i induced by malathion In Ca^2+^ medium, malathion-induced ↑ [Ca^2+^]i was inhibited by a PKC inhibitor	[45]
SH-SY5Y	Mipafox, paraoxon, fenamiphos, profhenophs: 1 × 10^−10^–1 × 10^−2^ M for 24 or 48 h	Mipafox induced calpain activation after 24 h	[41]
Snail neurons	Paraoxon: 0.3–0.6 μM for 10 min	- Modulation of PKC activity modified Ca^2+^ action potentials and neuronal activity, but did not contribute to the neurotoxic actions of paraoxon - TMB-8, an IP_3_ receptor-mediated intracellular Ca^2+^ release antagonist, suppressed the paraoxon-induced secondary increase in the duration of PHP and neuronal silencing	[58]

**Abbreviations:** DLT, deltamethrin; PND, postnatal day; S100*β*, S100 calcium-binding protein beta; PMCA, plasma membrane Ca^2+^-ATPase; CAMKII, calcium/calmodulin-dependent protein kinase II; mTOR, mammalian target of rapamycin; PLC, phospholipase C; PKC, protein kinase C; IP_3_, inositol 1,4,5-trisphosphate; PHP, posthyperpolarization.

**Table 5 ijms-22-13376-t005:** Proposed treatments to reverse the pesticide-induced alterations in Ca^2+^ homeostasis.

Species or Cellular Line	Dose and Time of Exposure	Results	Reference
Wistar rat	Carbofuran: 1 mg/kg oral for 28 days	NAC had a beneficial effect on Ca^2+^ homeostasis	[47]
Rat CGN cells	Rotenone: 2–50 nM for 30 min or 12 h	Creatine attenuated early rotenone-induced [Ca^2+^]i dysregulation	[42]
PC12 cells	DLT: 10 μM for 1 h	Tert-butylhydroquinone reduced the ↑ [Ca^2+^]i induced by DLT	[48]
PC12 cells and rat primary cortical cells	Rotenone: 0–1 μM for 24 h	Chelation of the [Ca^2+^]i with BAPTA-AM or prevention of extracellular Ca^2+^ entry by EGTA ↓ H_2_O_2_ overproduction	[49]
GHA and human glioblastoma DBTRG-05MG cells, and D1 TNC1 rat astrocytes	Malathion: 5–25 μM	Chelation of cytosolic Ca^2+^ with BAPTA-AM prevented the malathion-induced cytotoxicity	[45]
GHA and D1 TNC1 cells	LCT: 10–15 μM	Chelation of cytosolic Ca^2+^ with BAPTA-AM ↓ the apoptosis induced by LCT	[44]
Neuronal soma of land snail (*Caucasotachea atrolabiata*)	Paraoxon: 0.3 μM for 5 or 10 min	BAPTA-AM ↓ the duration and amplitude of PHP and ↑ the duration and frequency of Ca^2+^ peaks In the presence of BAPTA-AM, paraoxon ↓ the duration of Ca^2+^ peaks without affecting their frequency	[59]

**Abbreviations:** NAC, N-acetylcysteine; CGN, cerebellar granule neurons; DLT, deltamethrin; BAPTA-AM, 1,2-bis-(2-aminophenoxy)ethane-N,N,N′,N′-tetraacetic acid acetoxymethyl ester; GHA, Gibco^®^ human astrocytes; LCT, lambda-cyhalothrin; PHP, posthyperpolarization.

## Data Availability

Not applicable.

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
