# Peer review of "Systematic Review of Calcium Channels and Intracellular Calcium Signaling: Relevance to Pesticide Neurotoxicity"

_ijms, 2021, doi:10.3390/ijms222413376_

Round 1
Reviewer 1 Report
Well done. Readership may be relatively small in terms of pesticides, but large in terms of potential toxicity to humans.
The reason for the high rating was precisely because the authors handled the calcium channel discussion so completely. If there were to be a negative comment, it would be the sometimes over-whelming use of abbreviations, but that said, careful reading uncovered no points of dispute in the calcium channel discussion. In my view, a job well done. The issue you may be looking for is: How can the toxicity of pesticides enter the human food chain and where would the authors expect the most vulnerable place to find an illness to manifest? Usually one look looks for toxicity in cell lines rapidly turning over such as bone marrow and skin, but the damage might be more permanent in slow turn over cell lines of the nervous system. The most likely organ to suffer a massive overdose is lung due to inspired air contaminated by a mechanical delivery system such as a low flying single engine plane.
Author Response
We are very grateful for the constructive comments of the referee in order to improve our manuscript, in this document we have tried to answer your questions.

Reviewer 2 Report
The review entitled Systematic Review of Calcium Channels and Intracellular Calcium Signaling: Relevance to Pesticide Neurotoxicity is well written and the Authors explain how pesticides may impair calcium influx and signaling causing ROS generation that might be associated with neurodegenerative diseases.
There are some points that should be better clarified:
- The increased intracellular calcium due to cellular stimuli is a process to activate calcium signal and regulate the biology of the cell. Is the abnormal calcium release (amplitude, oscillations etc.) that may generate cell damage leading to cell death. This concept should be reinforced, otherwise it seems that calcium elevation is always harmful.
- The effects of pesticides were evaluated in some vertebrates and insects mainly in in vitro systems. How is the possible impact on human population?
- The toxic effect of pesticides supposedly occurs by respiratory way and polluted food ingestion. Is it known if the residual amounts of pesticides found in food are sufficient to induce neurodegeneration? Are there data from different world areas?
- Authors suggest that pesticides might be associated with neurodegenerative disorders such as Alzheimer and Huntington disease. Is not well clear which mechanism links pesticide contamination with disease development/progression. Can pesticides affect the onset of brain cancers?
Author Response

(The authors gave the same response as above.)

Reviewer 3 Report
In this Review, the authors aimed to provide detailed information on Calcium Channels and Intracellular Calcium Signaling and the Relevance to Pesticide Neurotoxicity
Comments and suggestions:
The introduction section contains too much data without too much understanding of the physiology of intracellular calcium. So for lines 21-52, it is recommended a scheme with their highlighting is useful for readers.
Tables contain too much written data, this should be a more concise summary of the written text. For example, authors may use arrows instead of up or down.
Figure 1 is blurred, the writing is not clear, it must have a higher resolution.
Consider revision accordingly!
Author Response

(The authors gave the same response as above.)

Round 2
Reviewer 3 Report
The authors addressed all my comments and the manuscript is ready for acceptance.